# First Identification and Molecular Characterization of *Trichinella britovi* (Nematoda: Trichinellidae) from the Pine Marten (*Martes martes* Linnaeus, 1758) in Romania

**DOI:** 10.3390/microorganisms11092339

**Published:** 2023-09-18

**Authors:** Ana-Maria Marin, Ovidiu-Alexandru Mederle, Gianluca Marucci, Dan-Cornel Popovici, Narcisa Mederle

**Affiliations:** 1Faculty of Veterinary Medicine, University of Life Sciences “King Michael I” from Timisoara, 300645 Timisoara, Romania; anamaria.marin@usvt.ro (A.-M.M.); narcisamederle@usvt.ro (N.M.); 2Department of Surgery, University of Medicine and Pharmacy “Victor Babes” from Timisoara, No. 2 Piaţa Eftimie Murgu, 300041 Timisoara, Romania; 3Department of Infectious Diseases, Istituto Superiore di Sanità, Viale Regina Elena 299, 00161 Rome, Italy; gianluca.marucci@iss.it; 4Forestry Faculty, Transilvania University Brasov, No. 1 Sirul Beethoven, 500123 Brasov, Romania; danpopovici30@yahoo.com

**Keywords:** *Trichinella britovi*, *Martes martes*, Romania

## Abstract

*Trichinella* spp. are etiological zoonotic agents spread throughout the world and affect mammals, birds, and reptiles; they evolve via two cycles: domestic and sylvatic. *Martes martes* is a carnivorous nocturnal mammal from the family Mustelidae. In Romania, this host is widespread in all forests of the country. *Martes martes* has an extremely voracious appetite, feeding on fruit and also on a variety of small animals, including rodents such as mice and rats. The aim of this study was the identification and molecular characterization of *Trichinella* larvae isolated from the muscle tissue of *Martes martes* collected in different counties of Romania. The muscle samples were examined via artificial digestion, and the larvae were identified at the species level via multiplex PCR. The presence of larvae belonging to *Trichinella britovi*, a species frequently identified in wild carnivores in temperate zones, was observed. Although *T. britovi* has been already reported in several host species in Romania, this is the first time this species has been observed in a *Martes martes* specimen. This finding contributes to our knowledge about the host species involved in the maintenance of the *Trichinella* sylvatic cycle in Romania, and it confirms that this parasite is consistently present in the wild fauna of this country.

## 1. Introduction

A large group of mammals, the mustelids, have conquered the northern hemisphere (but they also have representatives in South America or the south of the African continent, developing specialized adaptations, typical for the habitats in which they live (arboreal, aquatic, etc.) [1]. In Romania, there are many species of mustelids, including the badger (*Meles meles* Linnaeus, 1758), the European mink (*Mustela lutreola* Linnaeus, 1761), the European polecat (*Mustela putorius* Linnaeus, 1758), the least weasel (*Mustela nivalis* Linnaeus, 1766), the stoat (*Mustela erminea* Linnaeus, 1758), the otter (*Lutra lutra* Bonaparte, 1838), the stone marten (*Martes foina* Erxleben, 1777), and *Martes martes* [2]. *Martes martes* is present in all areas with forest vegetation, being found in high densities in deciduous or mixed forests in lowland and hill areas. An extremely active species, *Martes martes* does not shy away from leaving the canopy of trees, venturing onto the ground over kilometers in search of prey [3]. The food and habitat needs of pine marten, as well as the inter-relationships it has with other animals in the trophic pyramid, support its reservoir role and the involvement of this sylvatic animal in the transmission of various etiological agents, especially those with zoonotic potential (*Trichinella* spp.) [1].

*Trichinella* spp. are the etiological agents of a zoonosis that affects humans; they are transmitted by the consumption of raw or undercooked meat of animals infested with the larvae of these zoonotic nematodes [4]. Trichinellosis is a parasitic disease that affects mammals, birds, and carnivorous and omnivorous reptiles. The disease is widespread everywhere except Antarctica [5]. Currently, 13 taxa are described in the *Trichinella* genus; namely, the encapsulated species are *T. spiralis* [6], *T. nativa* [7], *T. britovi* [8], *T. murrelli* [9], *T. nelsoni* [7], *T. patagoniensis* [10], *T. chanchalensis* [11], and *Trichinella* genotypes T6, T8, and T9 (exclusively for mammals). The nonencapsulated species are *T. pseudospiralis* [12], *T. papuae* [13], and *T. zimbabwensis* [14], which can infect mammals and birds or mammals and reptiles [4].

In Romania, trichinellosis is a zoonosis with a high level of infection, and consumption of raw or undercooked pork but also of game meat (wild boar (*Sus scrofa* Linnaeus, 1758) and bear (*Ursus arctos* Linnaeus, 1758)) represents the major way of human infection [15,16].

In Europe, mustelids (*Meles meles*, *Martes foina*, etc.) and other carnivores (*Ursus arctos*, European lynx (*Lynx lynx* Linnaeus, 1758), wolf (*Canis lupus* Linnaeus, 1758), etc.) represent a source of infection with larvae of *Trichinella* spp. and have a role in the ecology of sylvatic trichinellosis, although the red fox (*Vulpes vulpes* Linnaeus, 1758) and the raccoon dog (*Nyctereutes procyonoides* Gray, 1834) are considered the most important reservoir hosts [17]. The sylvatic cycle plays a relevant role in the transmission of this zoonotic disease, acting directly when infected raw or undercooked game meat (especially *Sus scrofa*) is consumed and also indirectly via the transmission of *Trichinella* strains from wild animals to domestic animals in noncontrolled rearing conditions (pigs raised in a yard) [18].

In Romania, there are no reports that support the possible consumption of meat of wild carnivores, but these carnivores appear as invasive species and suitable hosts for *Trichinella* spp. [15,16,19]. In Romania, no infection of *Martes martes* with larvae of *Trichinella* spp. has been reported so far [15], but the presence of respiratory tract parasites such as *Crenosoma vulpis* (Dujardin, 1844) and *C. petrowi* (Molin, 1861) has been observed [20]. The aim of the present study was the identification and molecular characterization of *Trichinella* larvae isolated from the muscles tissue of *Martes martes* collected in different counties of Romania.

## 2. Materials and Methods

### 2.1. The Target Hosts

*Martes martes* is a nocturnal carnivorous mammal belonging to the family Mustelidae, with reddish-brown fur and down that fades to yellow, a yellowish-white neck patch, and a black tail. It is widespread in Europe (including Romania and the Republic of Moldova), Asia Minor, northern Iraq and Iran, the Caucasus, and Western Siberia [3]. In Romania, it is widespread in all the forests of the country. It is more frequently found in the mountains, up to the limit of the forest vegetation; but it is also found on the plains and in the forests. A hunter par excellence with predominantly arboreal activity, *Martes martes* has an extremely voracious appetite, feeding on fruit and on a variety of small animals, including rodents such as mice or rats [1]. The characteristics of its habitat, food, and biology make it easier to understand the involvement of this sylvatic animal in the maintenance and transmissibility of various pathogens, especially parasites with zoonotic potential, such as the nematodes of the genus *Trichinella.*

### 2.2. Diagnostic Procedures

The research was carried out over a period of two years, on a total of 12 *Martes martes* (four males and eight females) aged between 1 and 2 years. The collected animals were found dead (road kill) or legally hunted (15 September–31 March) in five counties of Romania: Timiș, Arad, Caraș-Severin, Vâlcea, and Olt (Figure 1). The animals were examined at the Parasitic Diseases Clinic of the Faculty of Veterinary Medicine Timisoara/ULST.

About 30 g of muscle from the diaphragm and foreleg muscles was harvested from each animal and tested for the presence of *Trichinella* spp. larvae via the artificial digestion method according to the Commission Regulation (EC) no. 1375/2015 [21]. After artificial digestion, larvae were collected, counted, stored in 96% ethanol, and sent to the European Reference Laboratory for Parasites (EURLP) (Rome, Italy) for species identification via multiplex PCR [22]. Briefly, DNA that was purified from single larvae DNA was extracted from single larvae using a DNA IQ System kit (Promega, Madison, WI, USA) and a Tissue and Hair Extraction kit (Promega, USA). Five primer sets, targeting specific regions (expansion segment V, ITS1 and ITS2) of the ribosomal DNA repeats, were used in multiplex PCR to obtain a species-specific electrophoretic DNA banding pattern [23,24].

## 3. Results

One out of the 12 *Martes martes* specimens collected from Timiș County (lat. 46.009390, long. 20.846470) tested positive for the presence of *Trichinella* spp. larvae. The larval burden was estimated to be 1.12 larvae per gram (LPG) of muscle tissue. Ten *Trichinella* larvae were individually tested via multiplex PCR and identified as *T. britovi* (Figure 2).

## 4. Discussion

Comparing the results of this study with those from the literature, it can be stated that it is the first identification of the nematode *T. britovi* in *Martes martes* in Romania. The European reservoir for *Trichinella* spp. is made up of wildlife, with wild animals being the most important source of infection for domestic pigs (*Sus scrofa domesticus* Erxleben, 1777), which is the main source of infection in other animals (e.g., horses) and especially in humans [25].

An ecological model of the parasitic system of *Trichinella* spp., based on predation, necrophagy, and cannibalism, as the main ecological factors, was provided by Varlamova A.I. et al. [26]. *Vulpes vulpes* plays a major role in the accumulation and distribution of larvae and in maintaining a stable circulation of natural trichinellosis outbreaks. On a secondary level are *Canis lupus*, *Nyctereutes procyonoides*, *Meles meles*, *Martes martes*, and *Martes foina*. Domestic carnivores can be infected by wild predators [25]. A monitoring program of the sensitive host species (*Sus scrofa domesticus*) and also of the wild fauna in the monitored region, as well as the evaluation of the factors (habitat characteristics, of the wild host) that favor the circulation of *Trichinella* spp. in nature is fundamental in assessing the risk to domestic animals [25].

In Europe, *T. britovi* and *T. spiralis* are the most prevalent species isolated from wildlife. *Trichinella britovi* is more widespread than *T. spiralis*, with a different prevalence depending on the carnivore family, with the exception of mustelids, in which only *T. britovi* has been identified [25]. The same authors also support the existence of a domestic cycle maintained by *T. britovi* in countries where *Sus scrofa domesticus* are raised in backyards, and the nematode can easily be transmitted from wild to domestic animals (Bulgaria, Croatia, Estonia, France, Italy, Macedonia, Poland, Romania, Spain, and Ukraine) [25].

*Trichinella britovi* parasitize wild carnivorous mammals of the families Canidae, Felidae, Mustelidae, Ursidae, and Viverridae, living in the temperate regions of Europe, western and northern Asia, and west Africa [18]. In Europe, *T. britovi* was detected in 89% and 38%, respectively, of *Trichinella* isolates from carnivores and *Sus scrofa* [25]. *T. britovi* can also affect *Sus scrofa domesticus* populations and is the second species of *Trichinella* that can affect human health [27,28].

The identification of *T. britovi* in wild carnivores living in the vicinity of rural localities may represent a method of transmission of this zoonosis to species of wild fauna of hunting interest (*Sus scrofa*) and, through it, to humans [15,19,29].

*Trichinella britovi* is a cold-resistant species; *T. britovi* larvae can survive frozen in carnivore muscle for up to one year and in pig muscle for up to three weeks; in comparison, *T. spiralis* larvae do not survive more than a few hours or a few days [18,30].

A species with a pronounced arboreal character, *Martes martes* dominates its sylvatic habitat, where it frequently feeds on numerous species of mammals, including squirrel (*Sciurus vulgaris* Linnaeus, 1758) and European *edible dormouse* (*Glis glis* Linnaeus, 1766); birds such as the common blackbird (*Turdus merula* Linnaeus, 1758), the stock pigeon (*Columba oenas* Linnaeus, 1758), and collared pigeon (*Columba palumbus* Linnaeus, 1758); and reptiles or amphibians, but undertakes numerous incursions outside the forest ecosystem in order to supplement its food requirements and establish new hunting territories [1]. Thus, it can reach the vicinity of human settlements, animal farms, and grain warehouses, where, in addition to rats or mice, it feeds on domestic birds, causing economic damage and representing a potential risk of infection [2].

In Europe, *T. britovi* has been reported in numerous sylvatic hosts: *Vulpes vulpes* [25,31], jackal (*Canis aureus* Linnaeus, 1758) [25,32], *Nyctereutes procyonoides* [25,33], *Canis lupus* [25,34], wild cat (*Felis silvestris* Schreber, 1777) [25,35], *Lynx lynx* [25,36], *Meles meles* [25,37], *Martes foina* [25,38], *Lutra lutra* [25], European beaver (*Castor fiber* Linnaeus, 1758) [39], *Sus scrofa* [25,40,41], and *Ursus arctos* [25,42]. Therefore, the occurrence of *T. britovi* in wildlife in Europe is reported in a significant number of hosts, but information is scanty regarding *Martes martes*. In Serbia, Klun I. et al. [43] examined 469 wild animals including *Martes martes*; the *Trichinella* species identified were *T. britovi* and *T. spiralis*.

In Lithuania, Senutaitė J. and Grikienienė J. [44] identified *Trichinella* spp. in the muscles tissue of wild animals, with a high prevalence of infection in *Vulpes vulpes*, *Nyctereutes procyonoides*, and *Martes martes* [44]. Similarly, an extensive study performed in Bulgaria in wild animals identified the presence of *Trichinella* spp. in 26 host animals’ muscles tissue, with a high prevalence in *Canis lupus*, *Martes martes*, and *Vulpes vulpes* [45].

In Slovakia, an assessment of *Trichinella* spp. infection mentioned *Martes martes* as the most affected host, with a much higher prevalence than in *Vulpes vulpes*. This epidemiological aspect reinforces the synanthropic behavior of these animals and, finally, their role in the epidemiology of trichinellosis [46,47].

Epidemiological inquiries carried out in Latvia revealed a high prevalence of *Trichinella* spp. infection in sylvatic carnivorous mammals, which highlights that they are good indicators for assessing the risk of infection with this nematode. The predominant species was *T. britovi* isolated even from the muscles of *Martes martes* [48,49,50].

In Latvia, the results of a study compared with the situation in Lithuania (Kaunas region) showed that *Martes* species are frequent natural reservoirs for *Trichinella* zoonotic agents in both countries. *Martes foina* was more susceptible to infection than *Martes martes*, and the prevalence of *Trichinella* infection in Latvia was higher than in Lithuania. The most widespread species in Latvia and Lithuania (Kaunas region) was *T. britovi* [51,52].

Moskwa B. et al. [37] identified, for the first time in Poland, larvae of *T. britovi* in the *Martes martes* in 2012 [37]. The identification of *T. spiralis* in the raccoon (*Procyon lotor* Linnaeus, 1758) and *Martes martes* in Poland reinforces the claim that new invasive species of carnivorous mammals become reservoirs for *Trichinella* spp. and are responsible for maintaining the infection in food chains, including game and domestic animals intended for human consumption [53]. Four years later, Cybulska A. et al. [54] isolated *T. britovi* from the muscles tissue of martens, with a maximum intensity of larval distribution in the tongue muscles.

In the central Abruzzi region of Italy, Badagliacca P. et al. [38] identified *T. britovi* in six wildlife hosts: *Martes martes*, *Martes foina*, *Canis lupus*, *Vulpes vulpes*, *Felis silvestris*, and *Sus scrofa* [38]. In the Scandinavian countries, *T. nativa*, *T. britovi*, and *T. spiralis* were identified in the muscles of *Martes martes*, *Meles meles*, *Nyctereutes procyonoides*, and *Mustela lutreola* [55], and also in *Sus scrofa* and *Lynx lynx* [56]. The American marten (*Martes americana*, Turton 1806) from Québec (Canada) was parasitized with the species *T. spiralis* [57]. In Romania, *Trichinella* infection is present and widespread throughout the country, affecting wild animals significantly [15]. In Romania, two species circulate: *T. spiralis* and *T. britovi*. Animals affected by this parasitism are *Vulpes vulpes*, *Canis aureus*, *Mustela lutreola*, *Canis lupus*, *Felis silvestris*, *Lynx lynx*, *Martes foina*, *Meles meles*, *Mustela erminea*, *Mustela putorius*, *Ursus arctos*, and *Sus scrofa*. *T. britovi* is the most widespread species in the sylvatic cycle in Romania [15,16,58,59,60,61,62,63,64,65].

The fact that in the present study T. britovi was identified in Romania in Martes martes, along with other host species (*Vulpes vulpes*, *Canis aureus*, *Canis lupus*, *Felis silvestris*, *Lynx lynx*, *Meles meles*, *Martes foina*, *Mustela erminea*, *Mustela putorius*, *Mustela lutreola*, *Ursus arctos*, *and Sus scrofa*) further confirms the hypothesis that these host species represent an important vector of trichinellosis transmission in the sylvatic environment, with direct effects on species of hunting interest, in particular, Sus scrofa and Ursus arctos, posing the risk of human infection.

In European countries, cases of human trichinellosis as a result of infection with *T. britovi* through the consumption of *Sus scrofa* meat have been reported in Serbia [66,67,68], Portugal [69], Italy [70,71,72,73,74], Spain and Sweden [75,76], Greece [77], Iran and Turkey [78], France [79,80,81], and Slovakia [82,83]. Moreover, human infection with *T. britovi* through the consumption of other wild and domestic animals’ meat has been reported, including jackal meat [84], horse meat [73], domestic dog meat [85], and domestic pig meat in Spain [86,87], Argentina [88], Bulgaria [89], Sardinia [90], Corsica [91], and Slovakia [85].

In Romania, no cases of human trichinellosis with *T. britovi* have been reported; only cases with *T. spiralis* have been reported [92].

*Martes martes* fulfils the role of a true vector for the transmission of parasites (especially zoonotic ones), being a transit element between the wild ecosystem and the peripheral domestic one like *Sus scrofa*.

## 5. Conclusions

The present study reports the first case of *T. britovi* infection in *Martes martes* in Romania. This finding contributes to increasing our knowledge about the host species involved in the maintenance of the Trichinella sylvatic cycle in Romania and confirms that this parasite is present at a consistent level in the wild fauna in this country.

## Figures and Tables

**Figure 1 microorganisms-11-02339-f001:**
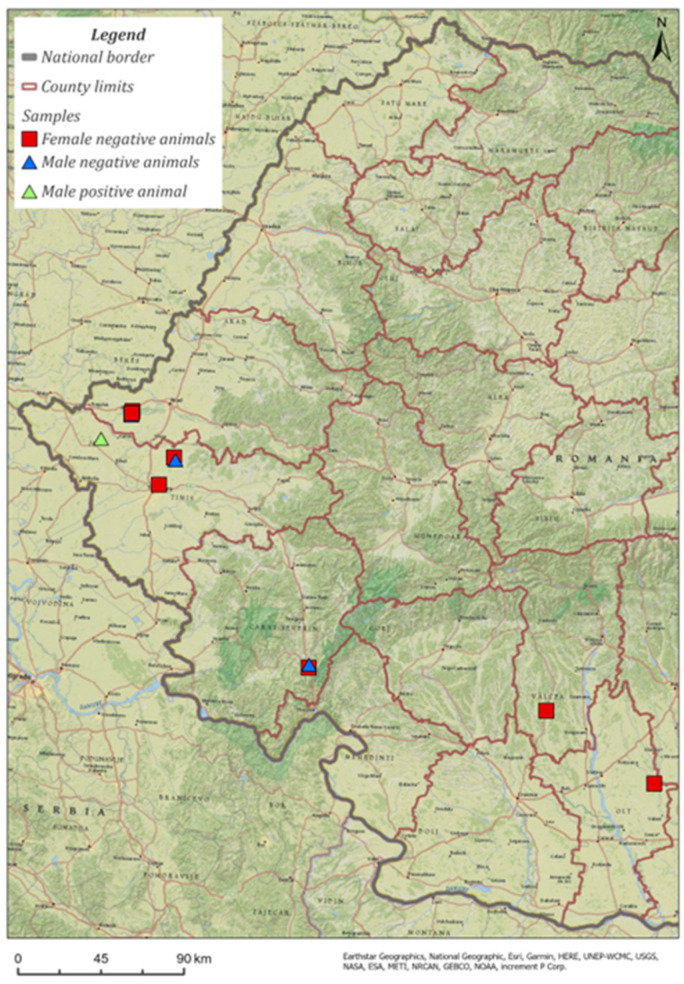
Map showing the geographical areas where *Martes martes* carcasses were collected; green triangle shows the sites here male positive animals were found.

**Figure 2 microorganisms-11-02339-f002:**
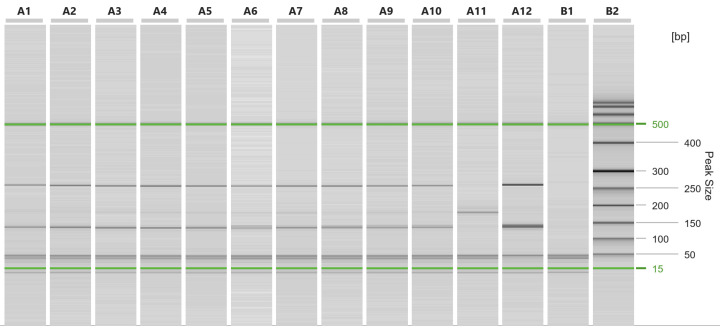
Capillary electrophoresis run of the multiplex PCR on *Trichinella* larvae collected from *Martes martes*. A1 to A10 larvae isolated from *Martes martes*; A11 *T. spiralis* positive control; A12 *T. britovi* positive control; B1 negative control; B2 size marker.

## Data Availability

Not applicable.

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
