# Peer review of "First Identification and Molecular Characterization of Trichinella britovi (Nematoda: Trichinellidae) from the Pine Marten (Martes martes Linnaeus, 1758) in Romania"

_microorganisms, 2023, doi:10.3390/microorganisms11092339_

Round 1
Reviewer 1 Report
Dear Authors,
Your research is very interesting.
I have some suggestions to improve their work.
The manuscript is attractive and well-designed; some significant points should be addressed before the manuscript can be considered for publication.
Keep the same font style and size throughout the manuscript and figures.
In methods, I suggest adding more information about the molecular identification of T. britovi, for example, the primer sequences, pairs of primers number, and DNA quantity.
Minor editing of English language required.
Minor editing of English language required.
Author Response
Response to Reviewer 1 Comments
Author Response to all reviewer comments: Thank you very much for your observations, they were very welcome and made us reorganize our paper. We upload the entire paper for you to reconsider for publishing as a Word document, where all corrections are suggested. We have used the track changes application, as suggested.
Dear Authors,
Your research is very interesting.
I have some suggestions to improve their work.
The manuscript is attractive and well-designed; some significant points should be addressed before the manuscript can be considered for publication.
Keep the same font style and size throughout the manuscript and figures.
Please find in the text the revision that you suggested
In methods, I suggest adding more information about the molecular identification of T. britovi, for example, the primer sequences, pairs of primers number, and DNA quantity.
Please find in the text the revision that you suggested
Minor editing of English language required.
Please find in the text the revision that you suggested
Reviewer 2 Report
The manuscript presents interesting data on the identification and molecular characterization of Trichinella larvae isolated from the muscles of martens from Romania. The study is well designed. In my opinion the results are correctly presented and the manuscript is well written.
However I have only one suggestion for the Authors before publishing the article in Microorganisms.
Line 13: Trichinella - please use the italics
Author Response
Response to Reviewer 2 Comments
Author Response to all reviewer comments: Thank you very much for your observations, they were very welcome and made us reorganize our paper. We upload the entire paper for you to reconsider for publishing as a Word document, where all corrections are suggested. We have used the track changes application, as suggested.
The manuscript presents interesting data on the identification and molecular characterization of Trichinella larvae isolated from the muscles of martens from Romania. The study is well designed.
In my opinion the results are correctly presented and the manuscript is well written.
However I have only one suggestion for the Authors before publishing the article in Microorganisms.
Line 13: Trichinella - please use the italics
Please find in the text the revision that you suggested
Reviewer 3 Report
The manuscript by Romanian colleagues deals with the first identification and molecular characterization of nematode Trichinella britovi in Martes martes from Romania. The article is interesting, is devoted to relevant topic. The article is sufficiently illustrated; its theme is relevant to Microorganisms.
However, I have some comments on this communication.
Regarding the title of the manuscript. Article “the” is missing (the Pine marten). In MDPI journals, in article titles, all words must begin with a capital letter. And I suggest changing the article name a little: “First Identification and Molecular Characterization of Trichinella britovi (Nematoda, Trichinellidae) from the Pine Marten (Martes martes) in Romania”.
The Introduction does not contain a formal purpose of the study. It is necessary to correct this as follows: “The aim of this study was …”
The legend in Figure 1 is not clearly visible. It should be included in the caption under Figure 1, as is customary in MDPI journals: “…Red triangles shows the finding sites of male positive animals…”, etc.
Regarding the writing of Latin names of species, orders, families, etc. Only Latin names of species and genera are written in italics. In the case of families and taxa above, italics are not used. For example, Trichinella spp. (Line 13), Mustelidae (Lines 15,76,136,137), etc.
My advice to authors: try to avoid common names of animals in scientific papers, use only Latin names (Martes martes) or common names must be used together with the Latin name (the Pine marten, Martes martes) and then only Latin. This applies to the entire text (Lines 19,26, 36, 38, 55, 57, 59,63,72,76,82,90,98,107, etc.).
At the first mention of all animal or parasite species, its full Latin name with the author and year of description should be given (as mentioned in the International Code of Zoological Nomenclature) (Lines 33-35, 49-52, 69,70). For example, the badger Meles meles (Linnaeus, 1758), Trichinella spiralis (Owen, 1835).
In the Introduction and throughout the text there are many paragraphs consisting of one sentence. This needs to be fixed. Such paragraphs must be appended above or below.
A one-sentence Сonclusion looks rather strange. Did the authors really come to only one conclusion from their study? Needs to be expanded. You can use abstract for this. And praise yourself J
The authors use the term “infestation” (Lines 61,174,180,199,209,210,213) throughout the text. But the term “infection” is more appropriate here. In any case, this is more common for me as a parasitologist.
The Communication is a short article, so you can include Figure from Supplementary Materials directly in the text.
Line 76 – “…to the family Mustelidae”.
Line 87 – Trichinella spp. – it's plural. That's why: “…the nematodes Trichinella spp.” or “...the nematodes of the genus Trichinella”.
Line 90 - Numbers up to 9 in articles are written in words – “…of two years…”
Lines 107-111 - Paragraphs need to be combined.
Lines 113-118, 125-128, 156-163, 183-184 - again paragraphs consisting of one sentence. You need to unite them.
Line 120,121 – correct reference – Varlamova et al. [15]. The initials of the authors are not included in the reference in the text.
Lines 130, 139, 145 - A sentence cannot begin with an abbreviation. Therefore, in such cases the Latin name is given in full.
Line 142 – missing point.
Lines 156-159 – Give the Latin species names. And if you want to keep the common names, don’t forget about the article.
Lines 162,164,183, 188, 191 - correct spelling to references. For example – Klun et al. [32].
Line 191 – delete year of publication.
The manuscript can be published, but corrections are needed.

English is understandable, however the manuscripts should be proofread by a professional translator or by a native speaker. There are problems with using articles and I would check the style of constructing English sentences.
Author Response

(The authors gave the same response as above.)
